# What Has Happened to Heartworm Disease in Europe in the Last 10 Years?

**DOI:** 10.3390/pathogens11091042

**Published:** 2022-09-13

**Authors:** Rodrigo Morchón, José Alberto Montoya-Alonso, Iván Rodríguez-Escolar, Elena Carretón

**Affiliations:** 1Zoonotic Diseases and One Health Group, Faculty of Pharmacy, University of Salamanca, Campus Miguel Unamuno, 37007 Salamanca, Spain; 2Internal Medicine, Faculty of Veterinary Medicine, University of Las Palmas de Gran Canaria, Campus Arucas, Arucas, 35413 Las Palmas, Spain

**Keywords:** *Dirofilaria immitis*, Europe, heartworm, epidemiology, dog, cat, wild animals, vector

## Abstract

Heartworm disease caused by *Dirofilaria immitis* is a vector-borne disease that affects canids and felids, both domestic and wild, throughout the world. It is a chronic disease which causes vascular damage in pulmonary arteries, and in advanced stages, the presence of pulmonary hypertension and right-sided congestive heart failure can be evidenced. Moreover, pulmonary thromboembolism is caused by the death of the worms, which can be lethal for the infected animal. Furthermore, it is the causative agent of human pulmonary dirofilariosis, being a zoonotic disease. The aim of this review was to update the current epidemiological situation of heartworm in Europe in dogs, cats, wild animals, and vectors insects, and to analyse the factors that may have contributed to the continuous spread of the disease in the last decade (2012–2021). In Europe, the disease has extended to eastern countries, being currently endemic in countries where previously only isolated or imported cases were reported. Furthermore, its prevalence has continued to increase in southern countries, traditionally endemic. This distribution trends and changes are influenced by several factors which are discussed in this review, such as the climate changes, presence of vectors in new areas, the appearance of new competent vector species in the continent, increased movement of pets that travelled to or originated from endemic countries, the urbanisation of rural areas leading to the formation of so-called “heat islands”, or the creation of extensive areas of irrigated crops. The continuous expansion of *D. immitis* must be monitored, and measures adapted to the situation of each country must be carried out for adequate control.

## 1. Introduction

*Dirofilaria immitis* is the causal agent of heartworm disease in domestic and wild carnivores, mainly dogs and cats. In addition, it is the causative agent of human pulmonary dirofilariosis, being a zoonotic disease. Its distribution is cosmopolitan, and currently, *D. immitis* can be found almost anywhere in the world, from tropical to cold climates. It is a vector-borne disease, in which the mosquitoes of genera *Culex* spp.; *Aedes* spp.; *Anopheles* spp.; *Coquillettidia* spp. act as vectors. When mosquitoes ingest blood with microfilariae, these moult to infective third stage larvae (L3) in approximately 14 days, depending on external temperature conditions: the higher the outside temperature, the shorter the moulting period. When the mosquitoes feed again, L3 are inoculated into the definitive host to continue their biological cycle [1,2].

Heartworm infection is a vascular and pulmonary disease. The dog is considered the main host of *D. immitis*, and it is characterised by the development of pulmonary endarteritis that chronically can lead to the development of pulmonary hypertension and right congestive heart failure, being a severe and potentially mortal condition (Venco et al.; 2008). In many cases, it is possible to find adult worms in the heart of infected dogs, and cases with >50 adult worms and microfilaremia in the blood in hyperendemic areas are a common finding [1]. However, in the feline host parasite, burdens are low, and one to three worms are generally found in the heart, most of them being amicrofilaremic. Cats are considered imperfect hosts and most infections are cleared by the feline immune system in their larval stages before they become adults [3]. For this reason, it is estimated that in endemic areas, such as Europe, the feline prevalence is between 5 and 20% of the canine prevalence in that same region [4]. Infected cats can present non-specific clinical signs, which may be transitory, or sometimes the sudden death of the animal is the only symptom shown [5,6]. For these reasons, the diagnosis of feline heartworm has traditionally been more difficult to achieve and, therefore, this disease has been less studied in this species.

As heartworm is a zoonotic disease, the presence of this parasite in canid and felid reservoirs poses a health risk to humans through the bite of infected mosquitoes. *D. immitis* can cause benign pulmonary nodules that can be mistaken for malignant lung tumours [1,7,8].

In Europe, the epidemiological situation in the two main hosts is markedly different. The number of studies addressing the epidemiological situation in dogs is significantly higher than in cats due to, among other factors, the difficulty in diagnosis and the number of cats compared to the number of dogs that can be accessed [2]. From 2011, heartworm disease was considered an endemic reality in eastern and central Europe, where it had spread from southern countries in a short period of time [1,9].

Currently, the epidemiological status has continued to change, as has the number of epidemiological reports. Europe is facing new climatic conditions caused by climate change, or the presence of new competent vectors that increase the time of exposure to the parasite (daytime and crepuscular/nocturnal mosquitoes) and that widen the risk zones (mosquitoes more resistant to low temperatures, overwintering eggs, etc.). This, along with other factors, has contributed to the establishment of vectors in regions where they were not previously found. Therefore, the objective of this manuscript is to review and analyse the current epidemiological situation in Europe during the last decade in dogs, cats, wild animals, and vectors to assess changes in the distribution of animal heartworm through a retrospective analysis of published epidemiological studies

## 2. Changes in the Distribution of Heartworm in Europe in the Last 10 Years (2012–2021)

### 2.1. Canine Heartworm

In the last 10 years, canine heartworm disease has continued to spread to eastern and north-eastern European countries, some of which are considered as new endemic countries. In other non-endemic countries, isolated cases or cases imported from endemic areas have continued to appear (Figure 1).

Moreover, in the traditionally endemic countries of southern Europe, prevalence in dogs has continued to increase, except for some territories where the control measures adopted have achieved a decrease in prevalence.

Spain and Portugal have traditionally been considered endemic countries, although not all regions of those countries have been studied [4]. In Portugal, the prevalence reported in the past has remained at similar percentages and has even increased in some regions. Cardoso et al. [10] carried out an extensive study analysing samples from all over the country and reported a general prevalence of 3.6%, obtaining prevalence of 2.9% in the north, 0.9% in the centre, 2.4% in the capital of Lisbon, and 5.1% in the south (Algarve). Furthermore, a prevalence of 9.4% was found in the coastal regions of the Algarve [11]. Other studies have reported prevalence of 15.1% in the central coastal regions of Portugal, while in the central region, prevalence ranged from 8.8 to 27.3% [12,13,14], 13.1% of dogs being microfilariemic in some studies [15]. According to Vieira et al. [13], the prevalence in the centre and northeast of the country was 2.1%.

In Spain, the studies have not been carried out uniformly throughout all territories, with several studies conducted in different regions in the past [1]. Recently, two nationwide studies have recently been carried out, reporting a general prevalence of 6.25–6.47% in the country [16,17]. In none of the autonomous communities analysed did the prevalence exceed 10%, except the Canary Islands (11.58%) and Balearic Islands (10.87%), being 5–10% in almost all regions and 1–5% in the northern communities. Moreover, heartworm is reported for the first time in Asturias, Cantabria, Navarra, and Basque Country, demonstrating the continuous spread of the infection in the whole country. The authors point out that the heterogeneous prevalence obtained in the different parts of the Iberian Peninsula, as well as in the Canary Islands, the Balearic Islands, and the cities of Ceuta and Melilla are related with the different bioclimatic factors in their different regions, such as temperature, humidity, orography, and vegetation. This heterogenicity can be observed when the prevalence was analysed by provinces, reporting high prevalence in the northwest (Pontevedra, 12.61%), south (Cádiz, 13.68%), Balearic Islands (Ibiza, 17.09%), or Canary Islands (Tenerife, 17.32%; Gran Canaria, 16.03%). In the Canary Islands, previous research has shown higher prevalence in the past, demonstrating that prevalence has gradually decreased over the years, probably due to prevention campaigns carried out by veterinarians [18]. However, disease is currently present in Lanzarote, previously considered free of heartworm [17]. Other studies carried out in northern and centre Spain have reported a prevalence of 0.18 and 7.19% for *D. immitis*, respectively [19,20]. Ad hoc studies have been carried out to analyse the prevalence in Barcelona, Madrid, Salamanca, and the Canary Islands. In the province of Barcelona, the average prevalence was 2.4%, being higher in municipalities belonging to the Baix Llobregat area, (11.1 to 33.3%), which has the highest risk of infection in the whole province [21]. The canine population in Madrid presented a prevalence of 3%, [22] and in the province of Salamanca, the overall percentage was 7.3% [20], although in zones with irrigated crops, the prevalence was higher. However, in the last years, the prevalence in these places has decreased from 29.08% to 16.7%, which could be attributed to the use of microfilaricides for the control of other parasitic diseases [23,24]. Finally, the cities of Ceuta and Melilla are studied for the first time (1.72 and 3.77%, respectively).

In Italy, epidemiological studies throughout the country on dogs have continued to be published. Previously, epidemiological reports indicated that prevalence in the north of the country was decreasing while in the centre and south of the country, it was increasing, and even regions with autochthonous cases were appearing in previously non-endemic areas [2,9,25,26,27,28]. Currently, there are also studies carried out in specific regions. For example, in Liguria (northwest), the presence of microfilaremic dogs was 0.6% and seroprevalence was 0.65%. In Friuli-Venezia Giulia, the prevalence was 5.9% [29]. In Padua (northeast), the prevalence of microfilaremic dogs was 18% [30,31]. In Latium, Tuscany, and Umbria (central Italy), prevalence ranged from 0.2 to 6.9% [29,32,33], while in southern Italy, the prevalence ranged between 0.2 and 44.2% [34,35,36,37]. In addition, Santoro et al. [38] reported two cases of heartworm disease in the Campania region of southern Italy. Finally, the prevalence in Giglio and Sardinia were 2.4 and 8.9%, respectively [29,39]. In addition, on the Linosa and Lampedusa islands, *D. immitis* was detected at 58.9% [40]. Finally, a study conducted by Mendoza-Roldan et al. [41] between 2009–2019 showed that the number of heartworm-infected dogs significantly varied over years, gradually increasing from 0.77% in 2009 to values ranging between 5.19–8.47% in 2016–2017. Moreover, the results showed that prevalence in the north is decreasing, while in the centre and south of the country, it is increasing, which corroborated studies previously carried out by other authors.

In France, while until 2011 no indication of expansion from the south, traditionally endemic, was published, there are recent data showing presence of canine heartworm in the centre of the country, with prevalence rates between 29.41–35.2% [9,42,43]. In addition, a case with microfilariae in blood was found in southeast France [44]. These studies indicated the high risk of infection in the central and southern regions of the country and the probable spread of the disease to the north. Finally, in Corsica, the prevalence was 21.3% [45], showing no changes with respect to previous studies [46].

In Austria, where previously only isolated cases were reported, there are already several studies reporting the solid presence of *D. immitis* in the country [47]. On the one hand, a study reported eighteen parasitized dogs with *D. immitis* and three with co-infections with *D. repens* [48]. On the other hand, there is a study analysing the presence of *D. immitis* from 1998 to 2018, which reported that the number of infections and co-infections with *D. repens* had increased significantly, especially from 2015 onwards [49], although most of the infected dogs had travelled to or originated from other countries (i.e., Hungary, Greece, Romania, Western Balkans, Spain, Portugal, Bulgaria, or the United States). Another recent study showed a high presence of heartworm in shelters from five different Austrian states (prevalence of 9.6%); although all infected dogs originated from Hungary, the authors warned about the risk of endemisation of *D. immitis* in Austria [50].

Germany faces a similar scenario, since it has been observed an increase in the number of infections in dogs that have travelled or came from endemic countries, but also in dogs that presumably have not travelled outside the country. Vrhovec et al. [51], after the examination of 30,970 and 54,103 canine blood or serum samples, reported a prevalence of 4.5% using Knott’s test and 1.4% using *D. immitis* antigens detection by ELISA. Another study reported a prevalence of 0.2% in the federal state of Brandenburg, in the centre of the country [52]. In addition, Schäfer et al. [53] reported five cases of dogs infected with *D. immitis* which had travelled to endemic countries (Spain, Greece, Hungary, and Italy). Finally, Maerz [54] reported 37 positive dogs diagnosed by clinical and imaging tests. This indicates that the parasite is invading new areas due to, among other factors, climate change and the expansion of the vectors that transmit the disease [55], so the status of Germany as an endemic or pre-endemic country should be considered.

In the United Kingdom and the Netherlands, Genchi et al. [56], based on the answers from veterinarians to a questionnaire in some European countries, have shown the existence of the disease in those non-endemic countries, considering the presence of *D. immitis* as isolated cases. In Ireland, for the first time, a clinical case has been published in a dog imported from the Canary Islands (Spain) [57].

In Russia, there is a study reporting a new endemic area, the Voronezh Natural Reserve, with prevalence ranging from 8.2 to 12.2% [58], which adds to the known endemic area of Rostov-na-Donu [59]. In Poland, the first native case of canine heartworm in the country is published [60], while a nationwide study reports a prevalence of 0.16% in the country [61]. In Ukraine, one infected dog has been reported in Kiev [62]. In Moldova, the first study on the occurrence of *D. immitis* has been published recently, demonstrating prevalence of 0.8% of microfilaremic dogs in Cahul and Chişinău [63].

Based on the answers to a questionnaire, autochthonous cases of *D. immitis* infections were seen by veterinarians in the Baltic countries, whereas most cases seen by veterinarians in the Nordic countries appeared to be imported [64]. Moreover, the first case of a microfilaremic dog in Lithuania, imported from an endemic area, was published [65].

Serbia and Croatia are considered endemic countries. In the latter, there are not reported increases in the prevalence, ranging from 0.46–0.6% in the published studies [66,67]. However, epidemiological studies from Serbia have shown increased prevalence, with ranges between 12.7 and 33.3% in the north of the country, together with the presence of some microfilaremic dogs [7,68,69,70,71]. In the Republic of Kosovo, antigens of *D. immitis* were detected in 14.8% of the dogs tested [72], being the first evidence of infected dogs in this country. In Bosnia and Herzegovina, there are no previous studies demonstrating the presence of *D. immitis* in dogs in these countries before 2011 [2]. In Bosnia and Herzegovina, the presence of *D. immitis* is reported for the first time, showing an overall prevalence of 1.89% in the region of the Bosnia-Podriuje canton (southern country), the prevalence being 0.93% for owned dogs and 2.86% for stray dogs [73].

Romania and Bulgaria are endemic countries as well, showing increased prevalence in the last decade. In Romania, reported prevalence has been increasing in correlation with increased interest and publications on canine heartworm. Mircean et al. [74] and Ionică et al. [75] reported average prevalence of 3.3 and 6.15%, respectively. In the eastern part of the country, the mean prevalence was 8.9%, half of the diagnosed animals being microfilariemic; moreover, some dogs presented *D. immitis* and *D. repens* co-infections. In the western and south-western regions, Giubega et al. [76] reported a prevalence of 9.74%, most of them microfilaremic. In Argeș County (centre/south of the country), the prevalence showed was 6.7% [77]. The highest prevalence was shown in Galati, in the southeast, 42.2% of the studied dogs being infected [78]. In another study carried out in dogs from four cities in Romania, *D. immitis* circulating antigens were found in 8.2% of the dogs; later, samples were heated and prevalence increased to 26.8% [79].

In Bulgaria, recent studies reported prevalence between 4.8–34.7%, being higher in the central, eastern, and southern regions and lower in the western and south-western regions [80,81,82]. In the central southern region, prevalence between 15.74 and 17.8% were reported, while prevalence in the south-western region was 5.5% [83,84]. Other studies have reported prevalence between 16.2 and 34.3% in the central-southern region (Stara Zagora) [85,86]. In the city of Sofia (western), the prevalence ranged between 4.8 and 25.8%, while the prevalence rose to 34.7% in the Sofia region [81,83,86,87,88]. In addition, Iliev et al. [80] showed a prevalence of 11.3% in the south and 8.3% in the north, with a mean prevalence of 10.5%. In another nationwide study, the percentage of positive dogs was 33% [89]. Finally, a positive dog was reported in the Pazardzhik region (southern Bulgaria) [82].

In Albania, *D. immitis* was present as isolated or imported in cases in the past [2], but recent studies demonstrated the endemic status of the country and a prevalence of 11.2% in police dogs from all over the country was reported in 2015 [90], while research in Tirana reported a prevalence of 2.2% [91]. In Greece, there has been an increase in the number of areas analysed in the recent years [4]. The reported prevalence in the urban cities of Thessaloniki, Larissa, Achaia, Attica, and Heraklion were 14, 7, 5.3, 0.7, and 0%, respectively. On the islands of Skiathos, Tinos, Ios, and Santorini, the reported prevalence was 0.5%. In all cases, both Knott’s test and serology were used to obtain the data [92,93]. Athanasiou et al. [94] carried out a study in which the average prevalence was 5.96%, 6.73% being in the north (Evos), 6.68% (Serres) and 6.16% (Florina), 7.3% in the northwest (Ioannina), 14% in the centre (Evritania), 5% in the centre-east (Evia), and 3.18% (Attiki). In another nationwide study, the prevalence obtained was 9.0% [95].

Turkey showed a similar trend, studies with prevalence of 1.5% being reported in Erzurum, north-eastern Turkey [96], 0.6% in Elazig Province [97], and a general prevalence of 8.5% in different provinces of Turkey distributed throughout the country [98]. These studies confirm higher prevalence than those reported 10 years ago [2]. In the Thrace Region of Turkey, the serologic and molecular assays determined a prevalence of *D. immitis* of 6.7 and 2.7%, respectively [99]. Finally, in different cities in the western country, such us İzmir, Aydin, Denizli, Mugla, and Manisa, an overall prevalence of 2.28% was reported [100]. In Cyprus, studies from 2019 reported for the first time the occurrence of *D. immitis* infections in dogs in the country, reporting prevalence of 0.5 and 4.3% [101,102], respectively.

The Czech Republic was considered endemic 10 years ago, with reports of autochthonous cases and infected dogs coming from neighbouring endemic countries [2,9]. However, in the last decade, there has been no increase in the number of reported cases and only one study has been published in a dog co-infected with *D. repens* in Příbram, in the Central Bohemian region, being an imported case of Hungary [103]. Until 10 years ago, in Slovakia and Hungary, reports of *D. immitis* were based on isolated or imported cases [2,4,9]. However, recent studies demonstrated the endemic status of these countries. In Slovakia, the number of studies has increased notably, evidencing the established presence of *D. immitis* in the country [47]. Two studies reported two infected dogs in eastern Slovakia and eight infected dogs in endemic areas of south-eastern and south-western Slovakia, all co-infected with *D. repens* [104,105]. In the same areas, Čabanová et al. [106] reported prevalence of 1.6 and 3.4%, respectively, while in the south-western of Slovakia the presence of sixteen symptomatic dogs was described [107]. In another study, an autochthonous case from a non-endemic region of south-eastern Slovakia was reported [108]. Finally, a prevalence of 3.26% of microfilaremic dogs and 1.4% of dogs co-infected with *D. repens* was reported in the western region, higher than in the eastern region, with no positive cases detected in the centre of the country [103]. In addition, two autochthonous cases of *D. immitis* infection were diagnosed in the Košice region of the southeast [108]. In Hungary, where the first autochthonous case was detected in 2009 [109], the number of epidemiological studies has increased considerably. There are studies that showed low-moderate prevalence between 0.36 and 8.1% [110,111,112,113] and co-infections with *D. repens* (3.2%) [111], while a high prevalence (51%) was reported in Szeged, a city close to Serbia and Romania [114]. Széll et al. [115] detected prevalence from 0.5 to 8% between 2007 and 2018, being most of the positive dogs located in the central, western, and southern regions of the country. In addition, an isolated case of a dog imported from Switzerland was described [116].

### 2.2. Feline Heartworm

The number of publications in feline heartworm is still substantially lower than in dogs. Some studies have reported *D. immitis* in cats in traditionally endemic countries, such as Spain, Portugal, Italy, and Greece, while the first reports of feline heartworm have been published in some new endemic countries as well (Germany and Austria).

In Spain, a recent nationwide study reported an overall seroprevalence of 9.4% (anti-*D. immitis* antibodies) and a prevalence of 0.5% (antigens) [117]. This study showed the highest seroprevalences in the Canary Islands (19.2%) and the Balearic Islands (16%), as well as the autonomous communities located on the Mediterranean coast (9.2–11.2%). Moreover, it is shown that the presence of this parasite is expanding throughout the national territory, also reporting infected cats in Ceuta and Melilla (Spanish autonomous cities on the north coast of Africa) and in the north of the country, similar to what has been reported for canine heartworm [117]. This study completed the epidemiological puzzle of several studies that had previously been carried out at a regional level, where the presence of cats infected by *D. immitis* had already been evidenced. It is the case of Zaragoza, where the seroprevalence reported was 24.4% and the prevalence was 3.6% [118]. In Madrid, seroprevalence and prevalence were 7.3 and 0.2%, respectively [22]. In Barcelona, 11.47% of the cats were seropositive and 0.26% were positive for *D. immitis* antigens [3]. Moreover, a previous study carried out in the Canary Islands demonstrated the hyperendemic status of the islands, showing a seroprevalence of 18.1%, although a remarkable disparity was found when evaluating the results by island separately, which ranged from 0% in El Hierro to 24.1% in Tenerife [18]. Moreover, on the island of Gran Canaria, a seroprevalence of 18.3% was recorded in cats in 2004 [119], so the results of the latest studies being 21.3% in 2016 [18] and 22.9% in 2022 [17] demonstrates a slow but constant increase in feline seroprevalences [120]. Furthermore, all studies concluded that the presence of feline seropositivity was linked to the presence of canine heartworm.

In Portugal, feline prevalence studies are relatively recent, but demonstrate the important presence of this parasite in the cats living in the country. The seroprevalence of samples collected in the centre of Portugal in 2009 and 2010 was 15% [13], the highest feline seroprevalences being found in Aveiro and Viseu (18.7 and 17.6%, respectively). Later, Maia et al. [121] reported a prevalence of 4.8% in southern Portugal. Moreover, a recent study reports for the first time a prevalence of 3.5% in cats on the island of Madeira [122].

In Italy, studies are still scarce. If previously, prevalence in cats was heterogeneous being 6.06–27% in the north and centre of the country, in addition to reports of isolated cases [4,9,123,124], in the last decade, studies in cats showed a prevalence of 17.6% in Linosa (an island halfway between the island of Sicily and the African coast of Tunisia) [40]. Furthermore, two cats co-infected with *D. repens* were reported in southern Italy [125] and isolated cases have been reported in different parts of the country [126,127,128]. In Greece, the first feline study was carried out in the north of the country, reporting a prevalence of 3% [93].

In Germany, a retrospective study from 2012 to 2020 identified heartworm in three cats in 2012 [129]. In Austria, the first case of autochthonous feline *D. immitis* infection is published in 2012 [130]. In Bulgaria and Romania, the first reports on feline heartworm infection are documented: in Bulgaria, a cat co-infected with *Ae. abstrusus* [131]; in Romania, a cat from Southern country [132].

### 2.3. Heartworm in Wild Carnivores

Although the studies focused on wild animals have increased substantially, they are still scarce. Between 2001 and 2011, a few studies were carried out in Italy, Spain, Portugal, and Bulgaria in foxes, otters, exotic felids, wolves, and jackals [2]. In the last ten years, Serbia was the country where most studies have been carried out, reporting naturally occurring patent infection with *D. immitis* in otters in areas near the Danube River in eastern regions of the country. References [133,134,135,136] reported *D. immitis* infection in a grey wolf and in a red fox. Penezić et al. [137] and Potkonjac et al. [134] showed presence of heartworm in golden jackals (prevalence of 7.32 and 3.1%, respectively), and in red foxes (1.55 and 5.3%, respectively). In addition, Penezić et al. [137] showed prevalence of 1.43% in wolves and 7.69% in wild cats.

In Hungary, a prevalence of 3.7% in red foxes and 7.4% in golden jackals was reported [112]. Moreover, an autochthonous *D. immitis* infection was also published in a ferret [47]. In southern France, the presence of *D. immitis* was identified in two of ninety-three examined red foxes [138]. In Portugal, pinnipeds were reported as new heartworm hosts in Algarve, southern Portugal, with a prevalence of 43.8% by real-time PCR [139]. In Romania, prevalence between 7.58–9.26% was reported in golden jackals, 0.33% in red foxes, and 10% in feral cats [140,141]. In Russia, *D. immitis* was found in foxes, jackals, and raccoon dogs with a global prevalence of 23.4% in Krasnodar Krai, being the highest prevalence on the European continent in the last 20 years [142]. In Italy, the prevalence observed in wolves was 13.6%, finding from three to five adult worms in most cases. Another study in northern Italy found heartworms in 1.42% of the wolves studied [143]. In Bulgaria, Panayotova-Pencheva et al. [82] reported heartworm infection in a golden jackal and a red fox from Pazardzhik and Plovdiv regions, located in southern Bulgaria. In addition, several studies have shown a prevalence of *D. immitis* antigens between 3–44.4% in red foxes, and 4.4–58.7% in golden jackals; furthermore, isolated cases of infected red foxes and golden jackals were detected by necropsy [83,89,144].

### 2.4. Heartworm Vectors

Until 2011, vectors that showed to be potential transmitters of dirofilariosis were *Cx. theileri* in Portugal, *Cx. pipiens* in peninsular Spain, and *Cx. theileri* in the Canary Islands (Spain), *Cx. pipiens*, *Ae. albopictus*, *Ae. caspius*, *An. maculipennis,* and *Cq. richiardii* in Italy, and *Cx. pipiens* and *Ae. vexans* in Turkey [2]. However, the number of competent vectors has increased significantly in the last years and new species of mosquitos able to transmit *D. immitis* have been identified. Moreover, the number of countries that have detected competent mosquitoes has also increased notably (Table 1).

In Portugal, the presence of *D. immitis* larvae were identified in new species of mosquitoes, such us *Cx. pipiens* s.l.; *Cx. theileri*, *Ae. caspius*, *An. maculipennis* s.l., and *Ae. dentritus* s.l. in Coimbra (centre), Santarém, and Setúbal (centre-southern) [145,146]. Similarly, *Cx. pipiens quinquefasciatus* was identified as a competent vector in Algarve [147]. In Spain, the existence of *Cx. pipiens* and *Cx. theileri* as potential vectors was reported in two endemic regions of the Iberian Peninsula [148,149]. In Italy, several studies have continued to demonstrate the existence of culicid vectors of *Dirofilaria* in different regions of the country, such as *Cx. pipiens* in north-eastern Italy [150], *Ae. albopictus, Cq. richiardii, Cs. annulate, Cx. pipiens,* and *Och. caspius* in southern Italy [34,151], and *Ae. albopictus* in Pelagie Archipelago [40]. Moreover, a new vector was detected in Europe, *Ae. koreicus*, which was identified in the Province of Belluno, north-eastern Italy, and in Chiasso, in the Swiss–Italian border region [152,153].

Mosquito species able to transmit the disease were reported for the first time in France, Belarus, Hungary, Germany, Moldova, Romania, Serbia, Slovakia, and Russia: in Corsica (France), *Cx. pipiens* and *Och. caspius* were identified as vectors for *D. immitis* [154]. In Germany, *D. immitis* larvae were detected in *Cx. pipiens/torrentium,* [155]. In the Southern Great Plain in Hungary, *Cx. modestus, Cx. pipiens,* and *Och. caspius* were identified as heartworm vectors; furthermore, *Cq. richiardii* was also identified in the country [156,157].

In Romania, in the Danube Delta Biosphere Reserve, Tomazatos et al. [158] reported a prevalence of 4.53% of infection by *D. immitis* in three species of mosquitoes; the highest prevalence was detected in *An. maculipennis*, followed by *Cq. richiardii* and *An. hyrcanus*. Other authors have reported the presence of heartworm larvae in the thorax/head of *Cx. pipiens* as well [141]. In Moldova, *An. maculipennis*, *Cx. pipiens/torrentium*, *Aedes behningi, and Anopheles pseudopictus* were *identified as vectors for the transmission of D. immitis* [159]. In Belarus, a country with no data on the occurrence of *D. immitis* in animals, *Cx. pipiens/torrentium* was identified as a vector of heartworm disease [159]. In Slovakia, two studies have demonstrated the presence of *D. immitis* larvae in *Ae. vexans* mosquitoes in eastern Slovakia [160] and *Cq. richiardii, Cx. pipiens pipiens,* and *Och. sticus* in western Slovakia [161]. In Serbia, the species reported as vectors were *Cx. pipens*, *Cq. richiardii,* and *Och. caspius* [162].

In Russia, Shaikevich et al. [163] estimated that the infection rates in mosquitoes from the central part of Russia varied from 3.1 to 3.7% and, in the southern region, from 1.1 to 3.0%. Moreover, the vector species in which *D. immitis* larvae were evident in the thorax were: *Ae. cantans, Ae. cataphylla, Ae. communis, Ae. intrudens, Ae. geniculatus, Ae. albopictus Cq. richiardii, Cx. modestus,* and *Cx. pipiens*; larvae were found in the abdomen of *Ae. cexans, Ae. cinereus, Ae. excrucians,* and *An. messeae*. In addition, in the Tula region near Moscow, Belarus, and Ukraine, the prevalence of *D. immitis* in infected mosquitoes was 1.5%, and larvae were found in *Ae. geniculatus*, *Ae. vexans, Cx. pipiens,* and *Och. Cantans* [164].

The presence and activity of vectors in a given region is directly related to climatic conditions. This is evident in the time needed for the larvae to develop to L3, which is temperature-dependent (8–10 days at 28–30 °C, 11–12 days at 24 °C, and 16–20 days at 22 °C), while below 14 °C, larvae development stops [165]. In this sense, in Europe, climate change is playing a major role in the development and increase in the number of vector species in regions where previously no infected vectors were reported.

**Table 1 pathogens-11-01042-t001:** Heartworm vectors detected in Europe until 2011 and between 2012–2021. No data (-).

European Countries	Until 2011	2012–2021
Portugal [2,145,146,147]	*Cx. theileri*	*Ae. dentritus* *Ae. caspius* *An. maculipennis* *Cx. pipiens* *Cx. quinquefasciatus* *Cx. theileri*
Spain [2,148,149]	*Cx. pipiens* *Cx. theileri*	*Cx. pipiens*
Italy [2,34,150,151,152,153]	*Ae. albopictus* *Ae. caspius* *An. maculipennis* *Cq. richiardii* *Cx. pipiens*	*Ae. albopictus* *Ae. koreicus* *Cq. richiardii* *Cs. annulate* *Cx. pipiens* *Och. caspius*
France [154]	-	*Cx. pipiens*
Turkey [2]	*Ae. vexans* *Cx. pipiens*	-
Germany [155]	*-*	*Cx. pipiens/torrentium*
Serbia [162]	-	*Cq. richiardii* *Cx. pipens* *Och. caspius*
Slovakia [160,161]	-	*Ae. vexans* *Cq. richiardii* *Cx. pipiens pipiens* *Och. sticus*
Belarus [159]		*Cx. pipiens/torrentium*
Moldova [159]	-	*Ae. behningi* *An. maculipennis* *An. pseudopictus* *Cx. pipiens/torrentium*
Romania [141,158]	-	*An. hyrcanus* *An. maculipennis* *Cq. richiardii* *Cx. pipiens*
Hungary [156,157]	-	*Ae. vexans* *Cq. richiardii* *Cx. modestus* *Cx. pipiens* *Och. caspius*
Russia [163,164]	-	*Ae. albopictus* *Ae. cantans* *Ae. cataphylla* *Ae. cexans*, *Ae. cinereus* *Ae. communis* *Ae. excrucians* *Ae. geniculatus* *Ae. intrudens* *Ae. vexans* *Cq. richiardii*, *Cx. modestus* *Cx. pipiens* *An. messeae* *Och. cantans*

The increase in temperatures favours the transmission of the disease to animals and humans by allowing the annual periods of activity of the mosquitoes to be lengthened and the stages of development of the larvae to be shortened.

This has led to the creation of more optimal conditions that allow the spread of mosquitoes to other areas where previously the climatic conditions prevented their presence [1,55,78,166,167]. Hungary is an example of this, where increases in temperatures between 1–2 °C were reported and has presumably contributed to spread the presence of *D. immitis* from the south to the north of the country [115]. In addition, climate change (among other factors) has surely played an important role in the appearance of competent vectors in new regions and countries where their presence had not previously been reported [159,162,163].

In relation to some species of culicid mosquitoes that have been reported as vectors of the disease in some countries in the last 10 years or the replacement by other species and/or the absence of other species, this may be due to the absence of studies in some countries in the last decade and the small number of studies that have been carried out on the subject, among other factors. More studies are needed to know whether environmental conditions have influenced or are influencing this change.

## 3. Factors Contributing to the Spread of Heartworm

After analysing the epidemiological and sero-epidemiological data of dogs, cats, and wildlife, as well as the presence of culicid vectors, studies show that the number of infected animals and vectors continues to increase. Thus, reports of *D. immitis* have increased in countries where the presence of this parasite was already known, while heartworm cases have been confirmed in regions and countries that to date remained free of its presence. Therefore, during the last decade, it has been shown that the disease continues to spread to countries in the centre and northeast of Europe.

In addition to climate change, mentioned above, there are other factors that could also be directly influencing changes in the distribution and expansion of *D. immitis*. Several epidemiological reviews carried out in this regard [1,2,4,9,168] discussed some of these factors, such as the movement of microfilaremic reservoirs from endemic areas or travel of pets from heartworm-free regions to endemic areas. Similarly, the increased awareness of the disease by veterinarians, pet-owners, as well as public health and scientific personnel in the last years, has contributed to the development of more studies that evidence the expansion of *D. immitis*. Moreover, as previously indicated, the introduction of new competent vectors that quickly adapt to the environmental conditions of the European countries (i.e., *Ae. aegypti*, *Ae. abopictus,* or *Ae. koreicus*, among others) is of great concern. There are other factors of a more markedly anthropogenic nature, such as the urbanisation of rural areas, especially in areas close to irrigated crops or near rivers, marshes, or lakes. Buildings and asphalt retain the heat of the day, leading to the formation of so-called “heat islands”, which favour the creation of microclimates that support the development of *D. immitis* larvae in mosquitoes even during the colder months [22]. The lack of control measures in wild animals that act as reservoirs is also a factor to be considered, as well as the lack of chemoprophylaxis in dogs and cats that live in areas that are considered non-endemic. This, together with the lack of knowledge about this parasite by the health authorities in those countries, favours the spread and settlement of the heartworm all over Europe.

## 4. New Studies to Assess the Risk of Heartworm Infection in Europe

Geospatial analysis, geographical information systems, remote sensing, global positioning systems, and virtual globes have been a breakthrough in the study of vector-borne diseases, since they have made it possible to create maps that show the risk of infection. Currently, the use of these tools has grown in importance in the knowledge of zoonotic diseases because it allows for the evaluation of the risk of transmission to the human host [169,170,171,172].

In Europe, until 2011, some studies on heartworm disease have been carried out by using geospatial analysis and Geographical Information Systems (GIS) in Italy; specifically in Naples, in Campania (southern Italy), and in regions located in the centre of the country [27,173,174]. Furthermore, studies based on GIS carried out in the whole continent [167] showed that when climate data collected at several weather stations in a given area were favourable for the development of mosquitoes, a presence of positive dogs was found in those same locations. In addition, Mortarino et al. [27] estimated the annual number of *D. immitis* generations in mosquitoes and determined that transmission was limited to the summer months, mainly July and August.

Moreover, considering that larval development does not occur below the threshold temperature of around 14 °C, the number of heartworm generations was determined and a high risk of infection was estimated in southern Spain and Portugal, some areas of southern France, Corsica, Bosnia and Herzegovina, Turkey, Cyprus, the coast of Albania, several regions of northern and southern Italy, Naples, Sardinia and Sicily, Greece, some areas of Bulgaria, south-eastern Ukraine, and the region of Rostov-na-Donu in Russia. Furthermore, a very high risk of infection was determined in localised areas of southern Spain, two coastal areas of Sicily, southern Calabria, southern Turkey, Cyprus, and in islands of southern Greece. The region with the longest risk of infection was in Murcia (southern Spain), which ranged from March 21 to November 11. It was also suggested that if the current trend of climate change continues, *D. immitis* infection will spread to areas previously considered as free of infection [167,175,176].

In the last decade, several randomised studies were carried out in Spain, Romania, and Hungary that helped to understand the risk of infection in these countries. In Spain, a GIS model was developed to predict the risk of *D. immitis* transmission in the country, based on data of temperature, rainfall, and the location of irrigated crops, concluding that the highest risk of transmission was located in several areas from the centre, east, and south of the Iberian Peninsula, where moderate/high temperatures concurred with the presence of extensive irrigated crops; similarly, high risk of transmission was also found in some places of the Canary Islands and the Balearic Islands. On the other hand, the lowest risk of infection was predicted in the north of the country and in dry non-irrigated regions of the southeast [177]. Another study carried out in the province of Barcelona (northeast Mediterranean coast of Spain) showed that most of the heartworm-infected dogs were located in areas where the GIS model predicted a high risk of infection, such as the banks of the Llobregat river, coastal areas with irrigated crops or neighbourhoods close to parks, and green urban areas [21].

A study of similar characteristics was carried out in Romania [78], where a high risk of infection in the southeast and moderate in the centre of the country were demonstrated, due to favourable temperature and humidity for the presence of culicid mosquitoes. These regions were characterised by floodplains; moreover, in these regions, the highest canine prevalence was found, corroborating the validity of the prediction model. In addition, heartworm seemed to be expanding towards the northeast of the country, where climatic conditions also favoured the transmission of this parasite. In this study, the GIS model predicted that the longest risk of infection was in Bucharest, between May 21 and September 20.

In Germany, a study was carried out using data on average daily temperatures recorded by weather stations close to the locations where the presence of vectors infected with *D. immitis* was evidenced, confirming the suitability of the climate in certain German regions for the establishment of natural cycles of heartworm transmission [55], which elevated the risk of infection.

In Hungary, a study carried out between 1995 and 2018 reported a rapid spread of the disease, based on the geo-referencing of infected animals (dogs and golden jackals) and the detection of anomalous annual temperature data compared to the four previous years. The authors concluded that two main factors were favouring this spread: the increase in temperatures as well as the increase and expansion in the population of golden jackals, the most important reservoir of heartworm in the fauna of the Mediterranean Balkans. In addition, results showed that the climate of the Great Hungarian Plain was the most suitable for the establishment of *D. immitis,* where the highest prevalence of infected animals was also found [115].

## 5. Conclusions

In the last 10 years, considering the epidemiological data, the possible factors influencing transmission, and forecasting studies using GIS models in different European territories, it can be stated that animal heartworm has continued to spread to eastern and north-eastern European countries, some of which are currently considered new endemic countries. In other countries, isolated cases or cases imported from endemic areas have continued to appear and have even increased. Moreover, in the traditionally endemic countries of southern Europe, prevalence and seroprevalence in animals have continued to increase, except for some territories where the control measures adopted have achieved a decrease in prevalence. So, at present, most of the European continent could be considered endemic. Therefore, it is necessary to continue to conduct epidemiological and sero-epidemiological studies in animals and vectors, both in endemic and non-endemic countries, in order to monitor distribution trends of *D. immitis*. Similarly, necessary control measures are necessary to prevent further expansion and/or contribute to the decrease in prevalence in endemic areas, which is especially important given the zoonotic nature of the infection. To this aim, specific guidelines adapted to each country would be necessary, as recommended by the European Society of Dirofilariosis and Angyostrongilosis.

## Figures and Tables

**Figure 1 pathogens-11-01042-f001:**
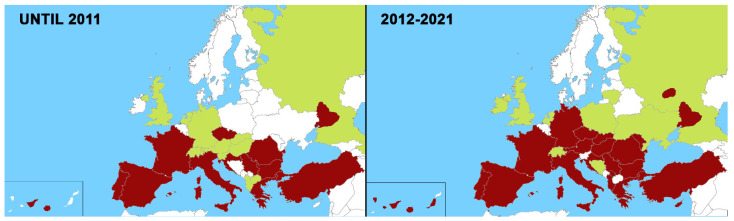
Geographical distribution of canine heartworm in Europe until 2011 and between 2012–2021. Endemic areas (🟥) and sporadic cases reported (🟨).

## Data Availability

Not applicable.

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
