# Peer review of "What Has Happened to Heartworm Disease in Europe in the Last 10 Years?"

_pathogens, 2022, doi:10.3390/pathogens11091042_

Round 1

Reviewer 1 Report

Pathogens-1871785

What has happened to heartworm disease in Europe in the last 10 years?

This is an interesting review of the heartworm epidemiology in Europe in the last 10 years. The information is updated and will be very useful for the monitoring the evolution of this disease in Europe.

Specific comments

Table 1. The information provided in this table, as it is presented, does not provide relevant data for the epidemiology of the disease. I suggest that the authors indicate the prevalence (ranges) of D. immitis reported by each country and indicate the bibliographic references within the same table.

Line 270. I suggest including a table similar to the one indicated in dogs (Table 1).

Line 316. If possible, include a table the same as the one indicated in dogs (Table 1).

Table 2. Indicate the reference within the table.

Conclusions

References should not be included in the conclusions. Figure 1 should be moved to section 4 (New studies to assess the risk of heartworm infection in Europe).

Author Response

All changes suggested by the reviewer have been added to the text and marked in red. In addition, the manuscript has been reviewed by a proofreading /editing service. The authors thank the referee for his dedication and time spent reviewing this article.

In relation to the specific comments, we have eliminated table 1 because it is repetitive with the map presented in figure 1, following your suggestion as it repeats information, and it is easier to see the change in the distribution in dogs in figure 1.

In relation to the rest of the suggestions regarding the inclusion of a new table, we believe that these would be repetitive tables of the text and would not provide clarity to the text. We believe that in its current state it would be good for the reader.

As for table 2 (now called Table 1) we have updated it with the references of the studies by country as suggested.

Regarding the Conclusions we have removed the errors you suggested we remove, and we have changed the place of Figure 1 by adding an introductory paragraph.

Reviewer 2 Report

Suggestions and comments to manuscript What has happened to heartworm disease in Europe in the last 10 years?”.

This is a very interesting MS on the current distribution of heartworm disease in different animal species, mainly dogs and cats living in many different European countries. I consider very interesting the inclusion of different infected wild species as well as the distribution of the different vector species of Diptera in the different European countries. Because of that, I consider this manuscript deserves published in Pathogens. However, I think some little mistakes should be solved prior to its publication.

General comments.

I suggest reviewing once more the use of English, since some inappropriate use of the language can be detected

Even though the authors describe the heartworm situation in detail country by country, some of these countries lack the epidemiological significance of that situation. It is interesting to show percentages of infections of dogs, cats and wild animals, but it would be better to show the epidemiological situation that this means in all the European countries studied, and not just in some. The same is true for vectors.

I consider it very important to explain the reason for the substitution of some vectors for others. It is interesting to know what species are currently present, however, I think it is necessary to explain the reason why some previously collected vectors were no longer recorded.

Please double check that all scientific names are correctly italicized

Minor changes.

Line 17. Please change to "vector insects", which is more descriptive.

 Lines 31-32. “Dirofilaria immitis is the causal agent of heartworm disease in domestic and wild carnivores, mainly dogs and cats.” Please rephrase. “Dirofilaria immitis is the causal agent of heartworm disease in wild (name main species) and domestic carnivores, mainly dogs and cats.”

Lines 44-45 “… mortal condition (Venco et al.; 2008).”. Please continue citing by number.

Line 58. Avoid starting a sentence with an abbreviation (“D.”)

Line 88. “….being microfilariemic 13.1% of dogs in some studies [15]”. Include those “studies” in addition to 15.

Lines 138-140. “Moreover, the results showed that prevalences in the north are decreasing, while in the centre and south of the country are increasing, which corroborated studies previously carried out by other authors”.  Please include the reasons for that variability.

Lines 271-274. “The number of publications in feline heartworm is still substantially lower than in dogs. Some studies have reported D. immitis in cats in traditionally endemic countries, such as Spain, Portugal, Italy and Greece, while the first reports of feline heartworm have been published in some new-endemic countries as well (Germany, Austria). A reference should be included.

Line 336. “….study in northern Italy found heartworms in 1.42% of the wolves studied (). In Bulgaria…”. This parenthesis seems out of place.

Author Response

All changes suggested by the reviewer have been added to the text and marked in red. In addition, the manuscript has been reviewed by a proofreading /editing service. The authors thank the referee for his dedication and time spent reviewing this article.

♠ Even though the authors describe the heartworm situation in detail country by country, some of these countries lack the epidemiological significance of that situation. It is interesting to show percentages of infections of dogs, cats and wild animals, but it would be better to show the epidemiological situation that this means in all the European countries studied, and not just in some. The same is true for vectors.

Thank you for your comment. In relation to this, in order to show the epidemiological situation, it would be necessary to make comprehensive studies of the disease area by area in each country where dogs, cats, wild animals, humans and vectors are analysed at the same time, in order to be able to compare data and, in addition, the same techniques have been used in all of them and, unfortunately, we are still far from that situation. We are currently carrying out a study in different areas of Europe to make a comprehensive study in Eastern, Central and Southern Europe. We have to complete with western countries, which is proving to be complicated.

♠ I consider it very important to explain the reason for the substitution of some vectors for others. It is interesting to know what species are currently present, however, I think it is necessary to explain the reason why some previously collected vectors were no longer recorded.

To clarify it, we introduce the following paragraph: In relation to some species of culicid mosquitoes that have been reported as vectors of the disease in some countries in the last 10 years or the replacement by other species and/or the absence of other species, this may be due to the absence of studies in some countries in the last decade and the small number of studies that have been carried out on the subject, among other factors. More studies are needed to know whether environmental conditions have influenced or are influencing this change.

♠ Please double check that all scientific names are correctly italicized

We have detected some errors and they have been corrected.

♠ I suggest reviewing once more the use of English, since some inappropriate use of the language can be detected

The manuscript has been reviewed by a proofreading /editing service

♠ Line 17. Please change to "vector insects", which is more descriptive.

OK. Thanks.

Round 2

Reviewer 1 Report

All the requests were adequately attended